# Association between the Immunophenotype of Peripheral Blood from mCRPC Patients and the Outcomes of Radium-223 Treatment

**DOI:** 10.3390/diagnostics13132222

**Published:** 2023-06-29

**Authors:** Elisabet Cantó, Georgia Anguera, Natalia Jiménez, Begoña Mellado, Ona Ramírez, Anais Mariscal, Pablo Maroto, Silvia Vidal

**Affiliations:** 1Inflammatory Diseases, Institut de Recerca de l’Hospital de la Santa Creu i Sant Pau, Biomedical Research Institute Sant Pau (IIB Sant Pau), 08041 Barcelona, Spain; svidal@santpau.cat; 2Department of Medical Oncology, Hospital de la Santa Creu i Sant Pau, Autonomous University of Barcelona, 08025 Barcelona, Spain; ganguera@santpau.cat (G.A.); jmaroto@santpau.cat (P.M.); 3Translational Genomics and Targeted Therapeutics in Solid Tumors Laboratory, Institut d’Investigacions Biomèdiques August Pi i Sunyer (IDIBAPS), 08036 Barcelona, Spain; najimenez@clinic.cat; 4Medical Oncology Department, Hospital Clinic Institut d’Investigacions Biomèdiques August Pi i Sunyer, 08036 Barcelona, Spain; bmellado@clinic.cat; 5Medical Oncology Department, Institut de Recerca de l’Hospital de la Santa Creu i Sant Pau, Biomedical Research Institute Sant Pau (IIB Sant Pau), 08041 Barcelona, Spain; oramirez@santpau.cat; 6Immunology Department, Hospital de la Santa Creu i Sant Pau, Biomedical Research Institute Sant Pau (IIB Sant Pau), 08041 Barcelona, Spain; amariscal@santpau.cat

**Keywords:** metastatic prostate cancer, radium-223, immunophenotype

## Abstract

(1) Background: Prostate cancer is the second most common cancer in men, with androgen suppression as the standard treatment. Despite initially responding to castration, most metastatic prostate cancer patients eventually experience progression. In these cases, Radium-223 is the chosen treatment. We hypothesized that the immunophenotype of circulating leukocytes conditions the response to Radium-223 treatment. (2) Material and Methods: In this prospective study, we collected peripheral blood from twelve mCRPC patients and nine healthy donors before (baseline) and during treatment with Radium-223. Immunophenotyping and the percentages of leukocyte–platelet complexes were determined by flow cytometry. The increments or decrements of leukocyte subsets between the baseline and the second Radium-223 injection were also calculated. (3) Results: At baseline, the mCRPC patients had a lower percentages of CD4^+^ T cells and B cells and higher percentages of NK and neutrophils than the HDs. In addition, they had more OX40^+^ CD4^+^ T cells, PD-L1^+^ CD8^low^ cells, PD-L1^+^ B cells, PD-L1^+^ NK cells, and monocyte–platelet complexes than the HDs. Moreover, patients with slow and fast progression had different percentages of PD-L1^+^ CD8^+^ T cells. In particular, slow progression patients underwent an increment of PD-L1^+^ CD8^+^ T cells after two cycles of Radium-223. (4) Conclusions: The characterization of circulating immune cells before initiating Radium-223 treatment could become a non-invasive indicator of the response.

## 1. Introduction

Prostate cancer (PC) is the second most common cancer in men and the fourth most common tumor type worldwide. It is an androgen-dependent tumor and most patients have a localized or regional disease. Androgen suppression (castration) is the standard treatment for metastatic diseases [1]. Most metastatic patients initially respond to castration. However, most of these cases may eventually progress and develop metastatic castration-resistant prostate cancer (mCRPC), with bone metastases localized in 70–80% of individuals [2]. This final evolutionary stage of the disease is incurable and associated with a significant risk of morbidity and mortality.

Radium-223 (Xofigo^®^) is an injectable chloride solution for the treatment of patients with mCRPC that have symptomatic bone metastases and no known visceral metastatic disease. Radium-223 is a first-in-class alpha-emitting radiopharmaceutical that has been shown to delay symptomatic skeletal events and prolong overall survival (OS) when compared to a placebo in the ALSYMPCA phase III clinical trial [3]. It provides a better quality of life for patients with mCRPC when compared to the placebo. Alpha particles cause irreparable cytotoxic DNA double-strand breaks in nearby tumor cells, osteoblasts, and osteoclasts. They have a shorter range, spanning 2–10 cell diameters, and a higher linear energy transfer, resulting in a highly specific effect with limited hematological toxicity [4].

In addition to its direct antitumor activity (cytotoxic effect), an effect on the immune system has been proposed for Radium-223, namely a decrease in the frequency of PD-1^+^ effector memory CD8^+^ T cells during treatment [5]. Using peripheral mononuclear cells from mCRPC patients treated with Radium-223, Creemers et al. [6] observed an increased expression of ICOS and PD-L1, PD-1, and TIM-3, as well as an increased percentage of T regulatory cells and myeloid-derived suppressor cells. Similarly, a recent study showed that in the exosome transcriptome derived from the plasma of patients with CRPC and bone metastases, three of the top ten altered pathways were involved in the immune regulation and immune checkpoint activation which were associated with OS [7].

Although there exists limited data regarding the changes between baseline and post Radium-223 treatment in mCRPC patients, no study has been conducted to analyze the baseline immune state and its relation with the outcome of mCRPC patients. For this purpose, we first characterize immune modulation in our cohort of patients treated with Radium-223 and then analyze the differences in the outcomes to this treatment according to the patients’ baseline immune states. To achieve this goal, we immunophenotyped circulating immune cells from mCRPC patients before the initiation of Radium-223 treatment and we determined how the immune baseline state may influence the response to Radium-223. The immune baseline state was related to changes observed in the immune blood cell population during Radium-223 treatment, and may influence progression-free survival (PFS).

## 2. Materials and Methods

### 2.1. Patient’s Samples

In this prospective, non-interventional, single-arm, exploratory study, a total of 12 patients diagnosed with mCRPC (stage IV at the time of sample collection) were included. The study was conducted at two centers in Barcelona, Spain, (Hospital de la Santa Creu i Sant Pau and Hospital Clínic) between October 2019 and November 2021. All patients met the same inclusion criteria and received the same treatment regimen of Radium-223. Patients were eligible if they had histologically confirmed adenocarcinoma of the prostate and no visceral metastases, testosterone level ≤ 1.7 nmol/L or ≤50 ng/dL or bilateral orchiectomy, an Eastern Cooperative Oncology Group (ECOG) performance status score of 0 to 2, two or more bone metastasis, progressing mCRPC to one AR-targeting therapy and docetaxel, and a life expectancy of 12 weeks. Additional eligibility criteria included absolute neutrophil count (ANC) > 1500/μL, hemoglobin ≥ 10.0 g/dL and platelet count ≥ 100,000/μL (no transfusions were permitted seven days prior to patient inclusion), creatinine < 2.5 mg/dL, and bilirubin < 1.5 ULN. Patients had to be receiving bone health agents (bisphosphonate or other approved bone targeting therapy) on stable doses for at least four weeks prior to starting on Radium-223. Patients with visceral metastases and/or lymph nodes > 3 cm in the short axis diameter were excluded. Prior treatment with a bone-homing radiopharmaceutical (e.g., Radium, strontium, or samarium) was not permitted. Other exclusion criteria were brain or leptomeningeal metastases, other malignancies in <3 years, any other anticancer treatment except for luteinizing hormone-releasing hormone agonist, prostate adenocarcinoma with neuroendocrine features or oat cell carcinoma of the prostate. All patients provided written informed consent. Subjects were treated with the standard dose of Radium-223 dichloride 50 kBq (0.0014 mCi)/kg intravenously every four weeks and for six cycles. At baseline and after the second Radium-223 injection, blood was collected for immunophenotyping. Patients were followed up for one year after initiation of Radium-223 treatment. For the efficacy assessment, OS and progression-free survival (PFS) were considered. OS was defined as the time from the first dose received to the date of death by any cause. PFS was defined as the time from the first dose of Radium-223 to the date of confirmed radiographic and clinical progression as per clinician evaluation.

### 2.2. Flow Cytometry

Freshly peripheral blood samples from nine healthy donors (HDs) and twelve mCRPC patients were collected in heparin tubes at baseline (before Radium-223 treatment started) and two weeks after the second Radium-223 injection. Blood cells were stained with specific antibodies to analyze the distribution of leukocyte subpopulations and the expression of immunomodulatory molecules on these cells by flow cytometry. Briefly, 100 µL of whole blood was stained at room temperature in the dark for 15 min with a panel of antibodies against: CD4-Viogreen, CD8-Vioblue, CD20-APCVio770, CD16-PerCpVio700 (Miltenyi Biotec, Bergisch Gladbach, Cologne, Germany), and CD14-PeCy7 (BD Bioscience, Franklin Lakes, NJ, USA). With this panel, we were able to identify: CD4^+^ T, CD8^+^ T, CD8^low^ cells (CD8^low^ CD16^+^), NK cells (CD8-CD16^+^), B cells (CD20^+^), monocytes (CD14^+^), and neutrophils (CD14-CD16^+^). Leukocyte-platelet complexes were identified with an additional marker against CD41a-FITC (Immunotools, Gladiolenwg, Friesoythe, Germany). We also stained cells with antibodies against immunomodulatory molecules: PD-1-PE, PD-L1-PE, TIM3-APC, OX40-PE, and CD158-PE (KIR) (BioLegend, San Diego, CA, USA). After staining, red blood cells were lysed and leukocytes fixed using 2 mL of BD FACS lysing solution 1X (diluted in H_2_O) (BD Bioscience) for 10 min at room temperature, washed two times, and cells were resuspended in 200 µL of phosphate buffer solution. Samples were acquired with the MACSQuant Analyzer 10 flow cytometer (Miltenyi Biotec) and the analyses (the percentage of positive cells (%) and count/µL) for each population were obtained using FlowJo version X (FlowJo LLC, Ashland, OR, USA). The percentage of immunomodulatory molecules was analyzed after limit detection was determined by fluorescence minus one (FMO). In Appendix A, we show the gating strategy for the detection of the leukocyte subpopulations and the expression of the immunomodulatory molecules of a representative sample from a HD.

### 2.3. Statistical Analysis

The Kolmogorov–Smirnov test was used to analyze data with a normal distribution. To describe our population, numbers and percentages were used for qualitative variables, while the median (interquartile ranges, IQR) was calculated for ordinal and quantitative variables with an asymmetric distribution. Comparisons between groups were tested with the Student’s *t*-test or the Mann–Whitney test, according to a Gaussian distribution. Correlation analyses were carried out with Spearman correlations. The log-rank Mantel–Cox test was used to analyze differences in PD-L1^+^ expression on CD8^+^ cells during the follow-up period. A receiver operating characteristic (ROC) curve analysis was also performed to determine the most accurate diagnostic method to discriminate between slow and fast progression. All *p*-values were based on a two-sided hypothesis, and those under 0.05 were considered statistically significant. All of the analyses were performed using Graph Pad Prism 7 software.

## 3. Results

### 3.1. Demographic and Clinical Characteristics of mCRPC Patients at Baseline

In Table 1, we show the demographic, clinical, and blood parameters of 12 mCRPC patients at baseline and age, weight, and height matched HDs.

The mean time of PFS was 120 ± 78.85 days and eight of the patients died during the follow up (273.30 ± 103 days). Three patients completed the six cycles of Radium-223 treatment.

We compared the cell counts/µL and percentage (%) of circulating immune cells between the mCRPC patients and HDs (Table 2). The mCRPC patients had lower absolute numbers and percentages of CD4^+^ T and B cells than the HDs. Regarding CD8^+^ T cells, the mCRPC patients had lower cell counts/uL than the HDs, but a similar percentage. The percentages of NK cells and neutrophils were higher in the mCRPC patients than in the HDs.

The neutrophil to lymphocyte ratio (NLR) was also calculated, and we found a higher NLR in the mCRPC patients than in the HDs (4.69 (3.04–6.12) vs. 1.93 (1.32–2.39); *p* < 0.001).

### 3.2. Baseline Expression of Immunomodulatory Molecules on Leukocytes and Leukocyte–Platelet Complexes

In Table 3, we show the percentages of immunomodulatory molecules in leukocyte subpopulations from the mCRPC patients and HDs, which were determined as described in Material and Methods. We did not find differences in the percentages of PD-1 and TIM3 in leukocyte subpopulations from the mCRPC patients and HDs. However, we found higher percentages of PD-L1^+^ CD8^low^ cells, PD-L1^+^ B cells, PD-L1^+^ NK cells, and OX40^+^ CD4^+^ T cells, but a lower percentage of KIR^+^ CD8^low^ cells, in the mCRPC patients than in HDs.

A representative dot plot of those markers with significant differences between the mCRPC patients and HDs is shown in Figure 1A–C.

Based on our previous data suggesting the immunomodulatory role of leukocyte–platelet complexes in cancer disease [8,9], we also compared the percentage of leukocyte–platelet complexes in peripheral blood from the mCRPC patients and HDs. We found that mCRPC patients had a higher percentage of monocyte–platelet complexes than HDs, as shown in Table 4 and Figure 1D.

### 3.3. Relationship between Baseline Blood Immune Subpopulations, Leukocyte–Platelet Complexes and the Clinic Characteristics of mCRPC Patients

We intended to determine the relationship between baseline blood immune subpopulations, leukocyte–platelet complexes and the clinic characteristics of the mCRPC patients. In Figure 2A, we compared the blood immune subpopulations and leukocyte–platelet complexes with the stage of the disease at the moment of diagnosis in order to analyze if the stage at diagnosis may influence the blood cell composition at the time of sample collection. To this end, we grouped patients into two groups. Group 1 (*n* = 5) were patients at stage I (*n* = 2), II (*n* = 2) and III (*n* = 1) at the moment of diagnosis. Group 2 (*n* = 7) were patients at stage IV at the moment of diagnosis. We found that patients in Group 2 had higher levels of circulating B cells (*p* = 0.07), neutrophils (*p* = 0.005), and NLR (*p* = 0.005), but lower levels of neutrophil–platelet complexes (*p* = 0.048) than patients in Group 1 (Figure 2A).

When patients were stratified based on their Gleason score at the moment of sample collection (grouped as ≤8 and ≥9), we found that patients with a Gleason score of ≥9 had higher percentages of CD8^low^ cells (*p* = 0.016), B cells (*p* = 0.008), and PD-L1^+^ B cells (*p* = 0.048), and a lower percentage of PD1^+^ CD8^+^ T cells (*p* = 0.028) and PD1^+^ CD8^low^ cells (*p* = 0.048) than patients with a Gleason score of ≤8 (Figure 2B). Additionally, patients with a Gleason score of ≥9 had a higher NLR than patients with a Gleason score of ≤8 (12.70 (4.94–26.95) vs. 3.80 (2.88–5.34), respectively, *p* = 0.045). Regarding ECOG performance status at the moment of sample collection, patients with ECOG 2 had a lower percentage of KIR^+^ CD8^low^ cells (*p* = 0.048) and B cell-platelet complexes (*p* = 0.030) than patients with ECOG 1. No differences were observed when patients were grouped according to previously received treatments (Abiraterone, Enzalutamide or chemotherapy).

### 3.4. Circulating PD-L1^+^ CD8^+^ T Cells at Baseline Were Associated with Progression-Free Survival in mCRPC Patients

To analyze the association between circulating immune cells at baseline and clinical outcomes, we stratified patients based on PFS. Patients were identified as slow progression patients when they had PFS > 120 days (*n* = 7), whereas they were considered fast progression patients when they had PFS < 120 days (*n* = 5).

Slow progression patients had a lower percentage of PD-L1^+^ CD8^+^ T cells (*p* = 0.03) (Figure 3A) and a tendency towards lower levels of CD4^+^ T cell—platelet complexes and CD8^+^ T cell—platelet complexes than fast progression patients. We did not find differences in the percentages of leukocyte subpopulations, NLR and the other immunomodulatory molecules analyzed. No differences in clinical characteristics, previous treatments, or hematological and biochemical values were observed between the two groups of patients (Appendix A).

Next, we determined the changes (incremental or decremental) in leukocyte subpopulations, immunomodulatory molecules and leukocyte–platelet complexes as a consequence of treatment between baseline and after two cycles of Radium-223. As shown in Figure 3B, slow progression patients had increased percentages of PD-L1^+^ CD8^+^ T cells (*p* = 0.06) and showed a tendency towards incremental changes in PD-L1^+^ CD4^+^ T cells (*p* = 0.1) between baseline and after two cycles of Radium-223. In contrast, fast progression patients did not have higher percentages of PD-L1^+^ on T cells. No changes in the percentage of PD-L1^+^ on B cells, NK cells, monocytes, and neutrophils, were observed when comparing slow and fast progression patients (Appendix A). The increments or decrements in leukocytes subpopulations, PD-1^+^, OX40^+^, and KIR^+^ expression were comparable between slow and fast progression patients (Appendix A). Compared to slow progression patients, fast progression patients had a tendency towards decreased CD4^+^ T cells–platelet complexes, CD8^+^ T cells–platelet complexes and B cells–platelet complexes (Appendix A). Regardless of the rate of progression, we observed an increment in the amount of prostate-specific antigen (PSA) between baseline and after two cycles of Radium-223 (Appendix A).

To assess the prognostic value of PD-L1^+^ CD8^+^ T cells, we performed an ROC curve analysis. The area under the curve (AUC) was 0.886 (95% CI: 0.707–1.000; <0.0001), with an observed sensitivity of 1.000 and a specificity of 0.714. The staircase graph (Figure 3C) shows the percentage of PFS patients with a percentage of PD-L1^+^ CD8^+^ T cells <17 or PD-L1^+^ CD8^+^ T cells ≥ 17 (*p* = 0.049).

## 4. Discussion

We found that the phenotype and composition of circulating leukocytes from the mCRPC patients were different from those of the HDs. Baseline immune cell composition was also associated with the clinical characteristics of the patients (stage of disease at the moment of diagnosis, Gleason score and ECOG). This fact probably indicates the exhausted status of the immune system, which determines the capacity of cells to be activated. In addition, in our cohort of patients, we identified a subgroup of patients with different cell composition at baseline, which was associated with PFS during the follow up.

Before initiating Radium-223 treatment, the mCRPC patients had lower percentages of CD4^+^ T cells and B cells and higher percentages of NK cells and neutrophils than the HDs. These findings suggest that the immune response of these patients shifted towards the innate system. One consequence of that was higher NLR in the mCRPC patients than in the HDs. In line with our findings, previous studies have shown alterations in the composition of circulating immune cells from diverse cancer patients (including lung, ovarian, and pancreatic cancer patients). In particular, some authors have shown that patients with different types of metastatic tumors had a lower percentage of B cells than HDs [10]. Regarding prostate cancer patients, other studies have shown that they have lower lymphocyte counts [11] and higher percentages of NK CD56bright than HDs, but with an exhausted phenotype [12]. As expected, these features have an impact on the susceptibility of cells to apoptosis, proliferative capacity and the ability to elaborate select effector functions [13].

At baseline, we also observed a different expression of immunomodulatory molecules on circulating leukocytes in the patients and HDs. Compared with the HDs, mCRPC patients had a higher percentage of PD-L1^+^ on CD8^low^ cells, B cells, and NK cells. This phenotype resembles a pre-activated status and it may have limited the function of PD-1^+^ immune cells [14]. We also found an increased percentage of OX40^+^ on CD4^+^ T cells and a lower percentage of KIR^+^ on CD8^low^ cells from patients. The elevated percentage of OX40 may reflect an attempt by the immune system to fight cancer progression. OX40, a member of the tumor necrosis factor receptor superfamily, is part of a potent costimulatory pathway that can enhance T cell memory, proliferation, survival, and anti-tumor activity [15]. A higher percentage of OX40^+^ has been reported in tumor immune infiltrates from non-small cell lung cancer patients [16], and several trials using agonists of OX40 have been initiated in combination with other immunotherapies to activate an anti-tumor response [17].

We found differences in leukocyte subpopulations, immunomodulatory molecules, and NLR at baseline. This may be due to the disease itself or it may reflect previous treatments, as previously documented [18,19]. Although the time between diagnosis and collection of the blood sample varies between patients, we saw higher NLR in patients diagnosed at stage IV [20] and in patients with a high Gleason score [21]. Even though baseline NLR is considered to be a prognostic factor in Radium-223-treated patients [22,23,24], we could not associate NLR values with either PFS or overall survival. Discrepancies with other reports in this regard can be explained by the fact that, in our cohort, only 58% of the patients received five cycles of Radium-223. In the other cohorts, patients received complete cycles of Radium-223 because they had a less severe disease, ECOG 0, and were younger [25]. These results suggest that the results regarding treatment efficacy obtained in randomized controlled trials cannot easily be translated to daily practice due to patient selection.

We found that the mCRPC patients had a higher percentage of monocyte–platelet complexes than the HDs. Other authors have reported higher percentages of these complexes in lung cancer patients [26], but this is the first description for mCRPC patients. There is known to be a close relationship between platelets and cancer, probably because tumor cells can activate platelets [27]. We speculate that the binding of platelets to monocytes is comparable to the binding to tumor cells. In this case, the platelet–tumor complexes will facilitate metastasis [28,29]. Recently, specific circulating platelet-coated tumor cells have been identified in mCRPC patients. This parameter was a combined prognostic and predictive biomarker for taxane–platinum combinations [30]. Future studies are needed to decipher the relationship between monocyte–platelet complexes with tumor circulating cells–platelet complexes.

The disease in the entire cohort of patients progressed during the follow up even though they received Radium-223 therapy. However, disease in seven patients progressed slower than in the other five patients. Interestingly, we showed that the percentage of PD-L1^+^ on CD8^+^ T cells determined the rate of progression and the capacity of these cells to undergo activation during the follow up. Slow progression patients had lower percentage of PD-L1^+^ CD8^+^ T cells than fast progression patients. In this scenario, we speculate that cells from slow progression patients can be further activated, whereas fast progression patients had an exhausted phenotype. Two observations may support this feature of cells from slow progression patients: first, there was an increase in PD-L1^+^ CD8^+^ T cells between baseline and the second cycle of Radium-223; second, the percentage of leukocyte–platelet complexes between baseline and the second cycle of Radium-223 increased. It has been reported that the activation of T cells is accompanied by an increased expression of PD-L1 [31,32] and an increased percentage of lymphocyte–platelet complexes [33,34].

The observed changes in the immunophenotyping of leukocytes after initiation of Radium-223 suggest that this therapy might have an impact on circulating immune cells. Two previous studies using peripheral blood mononuclear cells from mCRPC patients support this suggestion. Joseph et al., showed that the frequency of PD-1 expressing effector memory CD8^+^ T cells decreased four weeks after the first dose of Radium-223 [5]. Creemers et al. showed that the fraction of CD4^+^ and CD8^+^ T cells expressing PD-1 and PD-L1 increased slightly after the sixth Radium-223 injection [6]. Despite the small cohort, the results of this pilot study suggest there is a crucial role for circulating immune cells. More importantly, immunophenotyping at baseline could become a non-invasive indicator of response to treatment. However, we are aware that this study has some limitations. First of all, there was a reduced number of patients, partly because Radium-223 is only available for the last stage of metastatic prostate cancer, reducing the number of patients eligible for enrolment. It is necessary to highlight that this kind of patient is extremely fragile. Consequently, 66.6% of them died during the follow up and most of them had an early termination of Radium-223 treatment. Therefore, no data post Radium-223 treatment was available as planned at the beginning of the study. Second, due to obvious ethical reasons, we could not include a control arm of mCRPC patients without receiving Radium-223 therapy. Therefore, it remains uncertain which observed immunological changes result from the Radium-223 treatment and which are an indirect consequence of disease progression. Despite these limitations, we have enrolled a homogeneous group of patients to achieve less dispersion of results to reach the conclusions shown in this work. Further research on a large scale and expanding the study to include the expression of immunomodulatory molecules in biopsies from mCRPC patients is warranted to validate our findings. Additionally, our results emphasize the crucial role of the host immune system in the clinical outcomes of mCRPC patients and highlight the importance of analyzing the immune blood status before initiating treatments to determine the most suitable treatment or combination of treatments for each patient.

## Figures and Tables

**Figure 1 diagnostics-13-02222-f001:**
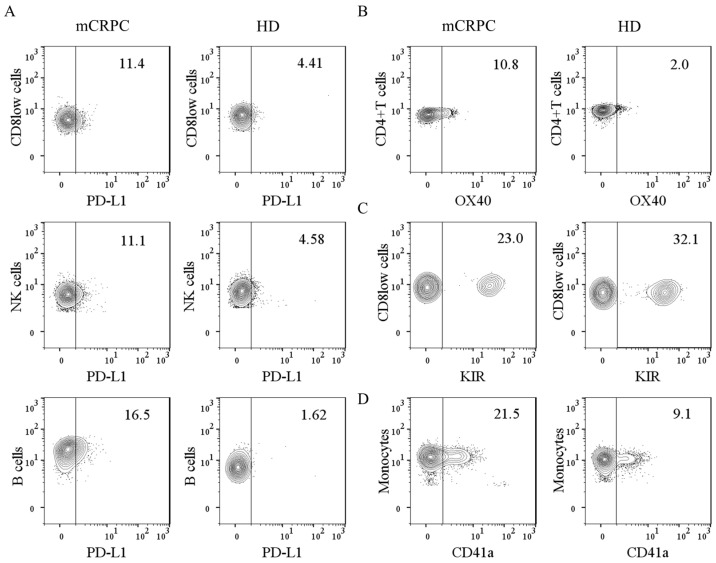
Differential expression of immunomodulatory molecules on blood leukocytes from mCRPC patients and HDs, determined by flow cytometry. As described in Material and Methods, whole blood were stained and the expression of immunomodulatory molecules were determined by flow cytometry. Representative dot plot is shown. (**A**) Expression of PD-L1, (**B**) OX40, (**C**) KIR and (**D**) monocyte-platelet complexes from mCRPC patients and HDs.

**Figure 2 diagnostics-13-02222-f002:**
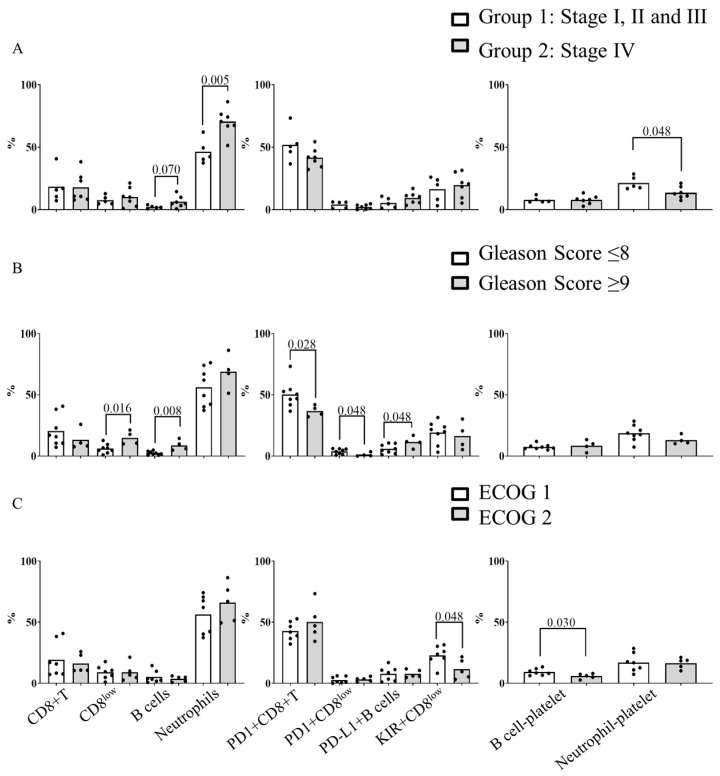
Relationship between blood circulating cells at baseline of mCRPC patients and clinical characteristics. Patients were classified accordingly: (**A**) stage at diagnosis. Group 1: stage I, II and III (white) and group 2: stage IV (grey); (**B**) patients with Gleason score ≤ 8 (white) and ≥9 (grey); and (**C**) patients with ECOG 1 (white) and ECOG 2 (grey). The Mann–Whitney test was used for the comparison of independent variables. *p*-values < 0.05 were considered significant.

**Figure 3 diagnostics-13-02222-f003:**
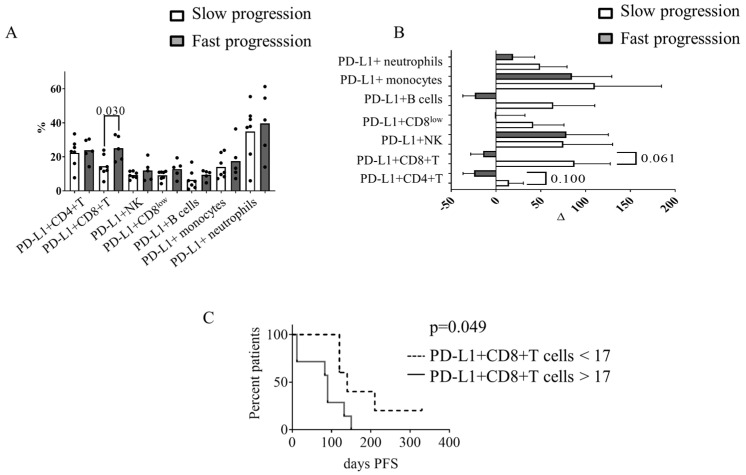
Circulating PD-L1^+^ CD8^+^ T cells at baseline were associated with PFS in mCRPC patients. Patients with slow progression in white (*n* = 7); patients with fast progression in grey (*n* = 5). (**A**) Percentage of PD-L1^+^ on leukocyte subpopulations at baseline; (**B**) increment or decrement in PD-L1 percentage between two weeks after the second Radium-223 injection and baseline; and (**C**) PFS during the follow up. The Mann–Whitney test was used for the comparison of independent variables. *p*-values < 0.05 were considered significant.

**Table 1 diagnostics-13-02222-t001:** Demographic, clinical and blood characteristics of mCRPC patients at baseline.

	Patients *n* = 12	HDs *n* = 9
Age	79.08 ± 7.62	62.48 ± 8.10
Weight (kg)	74.75 ± 10.10	68.13 ± 8.44
Height (cm)	168.18 ± 7.70	173.40 ± 3.92
Months between diagnosis and sample collection	94.10 ± 70.43	
Months between resistance to castration and sample collection	50.42 ± 37.78	
Month between metastasis and sample collection	47.82 ± 27.70	
Stage disease at moment of diagnosis (%) Stage disease at moment of sample collection	I (16%); II (16%); III (8%); IV (60%) IV (100%)	
Gleason score (%):		
≤8 ≥9 ECOG performance-status score (0–2): 1 2	66% 33% 7 5	
Median hematological and biochemical values:		
Platelets (×109/L)	217.89 ± 66.60	
Lymphocytes (×109/L) Neutrophils (×109/L) Monocytes (×109/L) Hemoglobin (g/L) Albumin (g/L) Alkaline phosphatase (U/L) PSA (µg/L) Lactate dehydrogenase (U/L)	1.32 ± 0.48 6.48 ± 3.58 0.50 ± 0.20 102.67 ± 14.47 38.85 ± 3.27 224.67 ± 197.39 305.77 ± 425.56 268.75 ± 69.22	
Previous treatment:		
Abiraterone (yes/no) Enzalutamide (yes/no) Chemotherapy (yes/no)	11/1 10/2 9/3	

Values are presented as mean ± SD. Gleason Score and ECOG were referred at the moment of sample collection. PSA: Prostate-specific antigen.

**Table 2 diagnostics-13-02222-t002:** Blood leukocyte composition in mCRPC patients at baseline and HDs.

	Count/µL		%	
	mCRPC	HD	mCRPC	HD
^a^ CD4^+^ T cells	**99.74 (53.27–160.8)**	**332.1 (237.5–380.5) *****	**35.61 (26.3–** **39.36)**	**40.14 (38.66–** **46.53) ****
^a^ CD8^+^ T cells	**43.98 (32.41–** **116.3)**	**164.4 (120.2–** **306.7) ****	16.45 (10.49–35.17)	22 (15.24–30.76)
^a^ CD8^low^ cells	18.98 (9.41–69.86)	31.8 (20.8–41.66)	8.17 (3.26–12.24)	3.77 (2.79–6.9)
^a^ B cells	**13.28 (4.19–** **32.01)**	**69 (43.76–** **104.5) ****	**3.60 (1.73–** **6.12)**	**8.68 (6.12–** **15.66) ****
^a^ NK cells	40.61 (27.52–134.3)	47.59 (21.7–97.58)	**14.58 (11.92–** **26.63)**	**6.45 (3.49–** **12.28) ****
Monocytes	109.5 (66.49–124.5)	165.5 (140.8–171.7)	94.23 (89.53–96.72)	94.52 (92.49–96.01)
Neutrophils	1734 (990.6–2382)	1489 (1038–1610)	**64.72 (44.09–** **73.20)**	**48.25 (45.52–** **53.07) ***

Data are presented as the median of the percentage (IQR: 25–75% percentile). * *p* < 0.05; ** *p* < 0.01; *** *p* < 0.001. ^a^ referred to lymphocytes. Bold typeset indicates significant difference.

**Table 3 diagnostics-13-02222-t003:** Expression of immunomodulatory molecules on blood leukocytes from mCRPC patients at baseline and HDs.

		PD-1	PD-L1	TIM3	OX40	KIR
^a^ CD4^+^ T cells	mCRPC	32.21 (27.92–44.28)	23.2 (17.58–29.15)		**8.58 (3.80–11.76) ****	
	HDs	24.65 (21.45–32.36)	17.35 (10.62–23.98)		**1.56 (1.09–3.73)**	
^a^ CD8^+^ T cells	mCRPC	44.56 (37.23–52.20)	17.85 (12.29–24.5)		0.29 (0.11–1.62)	0.86 (0.31–3.77)
	HDs	43.24 (30.2–51.49)	12 (8.21–20.88)		0.04 (0.025–0.15)	1.08 (0.47–2.74)
^a^ CD8^low^ cells	mCRPC	2.89 (0.82–5.49)	**10.86 (7.47–13.21) ***	3.17 (0.64–4.78)		**20.27 (8.56–25.44) ***
	HDs	5.69 (1.02–6.20)	**5.49 (3.61–9.51)**	2.01 (1.06–4.58)		**28.44 (25.36–30.53)**
^a^ B cells	mCRPC	1.32 (0.09–3.46)	**7.39 (3.75–11.08) ****		0.11 (0.01–0.20)	
	HDs	0.47 (0.21–0.67)	**1.74 (1.31–3.16)**		0.1 (0.03–0.18)	
^a^ NK cells	mCRPC	2.60 (2.33–4.47)	**9.69 (7.03–11.78) ***	4.89 (1.84–7.60)		17.71 (8.77–26.15)
	HDs	3.36 (1.82–14.5)	**5.91 (4.02–8.92)**	2.25 (1.80–6.2)		19.79 (15.89–27.09)
Monocytes	mCRPC		10.85 (7.47–17.95)	0.29 (0.13–0.78)		
	HDs		8.31 (5.84–9.16)	0.2 (0.08–0.3)		
Neutrophils	mCRPC		39.8 (22.9–51.87)			
	HDs		40.65 (30.4–47.58)			

Data are presented as the median of the percentage (IQR: 25–75% percentile). * *p* < 0.05; ** *p* < 0.01.^a^ referred to each population. Bold typeset indicates significant difference.

**Table 4 diagnostics-13-02222-t004:** Blood circulating leukocyte–platelet complexes in mCRPC at baseline and HDs.

	mCRPC	HD
CD4± platelet/CD4^+^ T cells	9.08 (7.41–10.62)	9.81 (8.35–11.67)
CD8± platelet/CD8^+^ T cells	8.26 (6.89–11.39)	9.42 (8.04–10.76)
CD8^low^−platelet/ CD8^low^ cells	7.88 (6.53–9.13)	9.14 (8.25–10.37)
B cells-platelet/B cells	7.69 (6.22–9.62)	9.5 (7.61–11.27)
NK cells-platelet/NK cells	8.00 (7.54–9.44)	11.19 (8.48–11.88)
Monocyte-platelet /Monocytes	**25.79 (18.03–34.68)**	**14.24 (10.75–19.55) ****
Neutrophil-platelet/Neutrophils	17.21 (11.34–20.54)	12.1 (9.23–14.8)

Data are presented as the median of the percentage (IQR: 25–75% percentile). ** *p* < 0.01. Bold typeset indicates significant difference.

## Data Availability

The data presented in this study are available on request from the corresponding author.

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
