# Peer review of "Association between the Immunophenotype of Peripheral Blood from mCRPC Patients and the Outcomes of Radium-223 Treatment"

_diagnostics, 2023, doi:10.3390/diagnostics13132222_

Round 1

Reviewer 1 Report (Previous Reviewer 1)

Elisabet et al's paper report a study that analyzed the association between the immunophenotype of peripheral blood from mCRPC patients and the outcomes of Radium-223 treatment. The authors collected blood samples from 12 mCRPC patients and nine healthy donors before and during Radium-223 treatment. They measured the percentages of different leukocyte subsets and leukocyte-platelet complexes by flow cytometry. They found that mCRPC patients had different immune profiles than healthy donors and that some immune markers were related to the progression and survival of mCRPC patients after Radium-223 treatment.The manuscript is well-written and well-organized. However, there are some areas that require improvement in the manuscript.

The study had a small sample size and a heterogeneous population of mCRPC patients, which may affect the statistical power and the validity of the results.

The study did not include a control group of mCRPC patients who did not receive Radium-223 treatment, which may limit the comparison of the immune changes induced by Radium-223.

The study did not measure the expression of immune checkpoints or cytokines in the tumor microenvironment, which may have a different impact on the antitumor response than the peripheral blood.

The study did not explore the potential synergies or interactions between Radium-223 and other treatments such as chemotherapy or immunotherapy, which may influence the outcomes of mCRPC patients.

Minor editing of English language required

Author Response

Reviewer 2 Report (Previous Reviewer 2)

Canto et al examines the immunophenotype of peripheral blood from metastatic castration-resistant prostate cancer (mCRPC) patients treated with Radium-223 treatment and compare it to the levels at the baseline to see if there is any correlation. To this end, the authors collect peripheral blood from 12 patients of different stages (??) and 9 healthy donors (HD). Then, different subsets of leukocytes were determined by flow cytometry between HD and mCRPC patients pre- and post-treatment. The authors report major differences in the leukocyte subsets of HD and mCRPC at the baseline level, prior to the Radium-223 treatment. Interestingly, the authors also claim a correlation between the percentage of PD-L1+CD8+T cells and the progression rate of the disease post-Radium 223 treatment. Based on this finding, the authors propose that circulating immune cells may be used as biomarkers to predict the effectiveness of Radium-223 treatment.

The data presented in this study could certainly be beneficial to the scientists and clinicians working in this field, especially with respect to the difference in the percentage of leukocytes between HD and mCRPC. However, the following issue on the patient number/type needs to be resolved before consideration for publication.

Major points:

1. Figure 2 is not displayed clearly; probably lost during conversion to PDF.

2. The authors state in Materials and Methods that the 12 patients were at Stage IV. However, they then state that 5 of them were at Stages I-II-III in the lines 285-286. It should be cleared at which stages the patients were at the beginning of the treatment. If they were all in Stage IV, then what is the point in presenting the data in Figure 2?

3. I wonder if the patients received the same treatment since they were treated at two different hospitals.

Minor points:

1. Please either use “Xofigo” or “Radium-223” to attain consistency

Moderate editing is required to increase clarity.

Author Response

This manuscript is a resubmission of an earlier submission. The following is a list of the peer review reports and author responses from that submission.

Round 1

Reviewer 1 Report

This article explores the hypothesis that the immunophenotype of circulating leukocytes influences the response to Radium-223 treatment in patients with metastatic castration-resistant prostate cancer (mCRPC). The study involved the collection of peripheral blood samples from 12 mCRPC patients and nine healthy donors, both before baseline and during Radium-223 treatment. Flow cytometry was utilized to perform immunophenotyping and determine the percentages of leukocyte-platelet complexes. The study also calculated the changes in leukocyte subsets between baseline and the second Radium-223 injection. The findings suggest that characterizing circulating immune cells before initiating Radium-223 treatment could serve as a non-invasive indicator of treatment response.

The manuscript is well-written and well-organized, and the data supports the conclusions. However, there are areas that require improvement. Firstly, it is essential for the authors to carefully review the text, eliminating any errors, such as the one found in line 41. Additionally, it is recommended that the authors refine their figures to ensure consistent font format, font size, and color scheme. This will enhance the appearance and readability of the figures.

To maintain consistency, it is advised to use a consistent name throughout the manuscript, such as "223-Ra" or "Radium-223". Ensuring the consistent use of terminology will help avoid confusion and enhance clarity for readers.

Minor editing of English language required

Reviewer 2 Report

Canto et al aims to examine the potential association between the immunophenotype of peripheral blood from metastatic castration-resistant prostate cancer (mCRPC) patients and the ramification(s) of Radium-223 treatment. To this end, the authors collect peripheral blood from 12 patients of different stages and 9 healthy donors (HD). Then, different subsets of leukocytes were determined by flow cytometry between HD and mCRPC patients pre- and post-treatment. The authors report major differences in the leukocyte subsets of HD and mCRPC prior to the Radium-223 treatment. Interestingly, the authors also claim a correlation between the percentage of PD-L1+CD8+T cells and the progression rate of the disease post-Radium 223 treatment. Based on this finding, the authors propose that circulating immune cells may be used as biomarkers to predict the effectiveness of Radium-223 treatment.

The data presented in this study could certainly be beneficial to the scientists and clinicians working in this field, especially the difference in the percentage of leukocytes between HD and mCRPC. However, I believe that the number of patients is not sufficient to claim the association between the percentage of circulating leukocytes and the effectiveness of the treatment, which is the major conclusion of the manuscript. Thus the current data does not warrant publication at its current form.

Major points:

  1. Line 69: I am not quite convinced with the statement that the authors “determined how the baseline immune state influences the response to....”. This statement is overstretching the results considering the low number of patients used in the study. I believe that the authors must examine more patients and specify the number of patients in each stage of the disease in Materials and Methods (Patient’s samples part).
  2. In Table 1, please provide the demographics of HDs as well (age..etc)
  3. Line 209: please specify the number of patients in each substage
  4. Figure 3. Please specify the patient number in the caption.
  5. Lines 331-334: Discuss the number of patients and the caution associated with the low number of samples. The same applies to the phrase “despite the limited cohort” in the line 349. Please discuss what should be the acceptable number of patients in a cohort.. etc.

Minor points:

  1. Line 41, please remove the bracket in the reference #1.
  2. Lines 62-64, This sentence could be combined with the previous paragraph.
  3. Line 102, please write out Healthy Donor before abbreviating it
  4. Line 114, “fixed” should be “were fixed”

Fine. Contains a few typos.

Round 2

Reviewer 2 Report

The authors have addressed all the points to their best ability. However, I still have reservations about the cohort size. For instance, in lines 262-266, the authors state that the study involed 1-5 patients in each stage of the disease at the moment of diagnosis. Accordingly, the data presented in Figure 2 come from such a low number of patients (one in stage 3 and 3 in stages 1 and 2).